# Depletion of Heavy Ion Abundances in Slow Solar Wind and Its Association with Quiet Sun Regions

**Liang Zhao \***[ID]**, Enrico Landi, Susan T. Lepri and Daniel Carpenter**[ID]

Department of Climate and Space Sciences and Engineering, University of Michigan, Ann Arbor, MI 48109, USA; elandi@umich.edu (E.L.); slepri@umich.edu (S.T.L.); dcar@umich.edu (D.C.)
* Correspondence: lzh@umich.edu

**Abstract:** The exact coronal origin of the slow-speed solar wind has been under debate for decades in the Heliophysics community. Besides the solar wind speed, the heavy ion composition, including the elemental abundances and charge state ratios, are widely used as diagnostic tool to investigate the coronal origins of the slow wind. In this study, we recognize a subset of slow speed solar wind that is located on the upper boundary of the data distribution in the $O^{7+}/O^{6+}$ versus $C^{6+}/C^{5+}$ plot (O-C plot). In addition, in this wind the elemental abundances relative to protons, such as N/P, O/P, Ne/P, Mg/P, Si/P, S/P, Fe/P, He/P, and C/P are systemically depleted. We compare these winds ("upper depleted wind" or UDW hereafter) with the slow winds that are located in the main stream of the O-C plot and possess comparable Carbon abundance range as the depletion wind ("normal-depletion-wind", or NDW hereafter). We find that the proton density in the UDW is about 27.5% lower than in the NDW. Charge state ratios of $O^{7+}/O^{6+}$, $O^{7+}/O$, and $O^{8+}/O$ are decreased by 64.4%, 54.5%, and 52.1%, respectively. The occurrence rate of these UDW is anti-correlated with solar cycle. By tracing the wind along PFSS field lines back to the Sun, we find that the coronal origins of the UDW are more likely associated with quiet Sun regions, while the NDW are mainly associated with active regions and HCS-streamer.

**Keywords:** solar wind; space plasma; solar corona; heavy ion composition

## 1. Introduction

The solar wind, continuous streams of ionized particles that are accelerated from the Sun, is crucial to space weather science and forecasting because its plasma properties largely determine the local conditions in the entire Heliosphere and the geo-effectiveness of Interplanetary Coronal Mass Ejections (ICMEs). In addition, the in-situ properties of the solar wind plasma are inextricably tied to the properties of the inner corona where it is heated, ionized and accelerated. Therefore, understanding the origins of the solar wind is important to understanding the key physics occurring in the Heliosphere, and to forecast Space Weather effects on the Earth.

Despite decades of study for the origin of the solar wind, the solar origin of the non-transient (ICMEs exclusive) slow-speed solar wind is still unclear. While, the fast-speed wind coronal origins are believed to be mainly polar coronal holes [1], and ICMEs that are mostly associated with solar eruptions, such as solar flares, etc., the slow-speed solar wind does not have a sole connection to a specific source region in the Sun, but rather it has multiple ones. The slow solar wind might be originating from anywhere outside polar coronal holes, including the periphery of active regions (e.g., [2,3]), helmet streamers [4–6], coronal hole boundaries [7], or pseudostreamers [8]. The latter structure consists of two sets of closed magnetic loops that separate open flux with the same polarity [9], and has been abundant between 2007 and 2009 [10,11]. The solar wind originating from pseudostreamer regions has been suggested to have slow to intermediate speeds and to be highly ionized (high $O^{7+}/O^{6+}$ ratio). Besides these origins, the slow-speed wind can also come from

low-latitude coronal holes. For example, slow solar wind (V < 600 km/s) observed by ACE around the ecliptic plane has been associated with low-latitude coronal holes because its variability, composition, and properties are very similar to coronal-hole-associated fast wind [4,6].

In order to bring insight into this long-lasting mystery of the origin of the slow-speed solar wind, we utilize the heavy ion composition measurements, because linking in-situ measurements to the wind source regions requires physical connections that allow us to associate any given plasma parcel detected in-situ to the inner solar atmosphere. Solar wind heavy ion composition measurements play such a role for two reasons.

First, the wind's ionization state reflects the plasma properties of the inner corona where the solar wind originates. As the solar wind is accelerated, the plasma ionization and recombination rates are proportional to the electron density, which rapidly decreases with the height above the Sun. At a critical height the electron density becomes so low that the two processes effectively stop [12] and the ionization state of an element remains unaltered as the solar wind propagates into the Heliosphere. This is the so-called "freeze-in" process. Beyond the freeze-in point (usually in the 1.5∼3.5 $R_s$ height range), the charge states of the heavy ions and their ratios (e.g., $O^{7+}/O^{6+}$, $C^{6+}/C^{5+}$) maintain a record of the thermal properties of the plasma in the inner corona they experienced below the freeze-in distance. Thus, in-situ measurements of heavy ion charge states can provide critical information of the thermal properties of the inner corona and can be used as effective diagnostic tools to study where the solar wind is originated (e.g., [1,4,5,13].)

Second, element abundance ratios can be affected by fractionation processes in the upper chromosphere resulting in a significant enhancement of the abundance of elements with low First Ionization Potential (FIP < 10 eV) in closed magnetic field structures over their values in photosphere [6,14–16]. This is the so-called FIP effect [17]. Such fractionation is directly inherited by the wind plasma and once the wind plasma is accelerated away, the abundance values remain unaltered and thus maintain record of the composition of the source region. This process provides us with one of the most important tools at our disposal to identify the coronal sources of different types of solar wind from in-situ data.

Since the Oxygen and Carbon charge state ratios freeze-in at similar heights, where theoretical models predict that the electron temperature of the corona monotonically increases before reaching its maximum, their charge state distributions are expected to be correlated. This correlation, such as the correlation between $O^{7+}/O^{6+}$ and $C^{6+}/C^{5+}$ ratios as measured by the Solar Wind Ion Composition Spectrometer (SWICS) on board ACE [18] has been investigated by the previous studies [13,19,20]. Interestingly, there are a subset of slow-speed wind whose largely depleted $C^{6+}/C^{5+}$ ratio make it be a group of outliers apart away from the main-stream correlation of the two charge state ratio plot ("O-C plot", hereafter, as Figure 1 left column). Further investigation shows that the systematic depletion of the fully stripped ions, such as $C^{6+}$, $O^{8+}$, $N^{7+}$, was the cause of this subset of slow solar wind ("Outlier" wind, hereafter) to be located in the right-bottom corner in the O-C plot. In addition, these Outliers are found to be more abundant during solar maximum and less during solar minimum, consistent with their solar origins that are associated with active regions [13], heliospheric current sheet and small magnetic flux ropes [20].

Besides the Outlier slow winds, whose peculiar feature and behavior can be characterized by the heavy ion composition, in this paper, we focus on another subset of slow-speed solar wind whose heavy ion abundances are systematically depleted compared to the normal slow wind. These winds are located on the upper boundary of the mainstream distribution of the O-C plot. We call these winds "Upper Depleted Wind" or UDW hereafter. We analyze the physical properties of these UDWs and investigate their coronal origins.

This paper is organized as follows: in Section 2, we describe the in-situ data that we analyze in this study and define the two groups of slow-speed solar wind that we compare, the UDW and those slow winds located in the main stream distribution of the O-C plot and possess comparable Carbon abundance range as UDW ("normal-depletion-wind", or NDW hereafter). In Section 3, we compare the in-situ properties and coronal

origins of the UDW and NDW. Section 4 summarizes the results. In Section 5 we draw our conclusions and discuss the implication of the UDW's features in light of possible solar wind acceleration process.

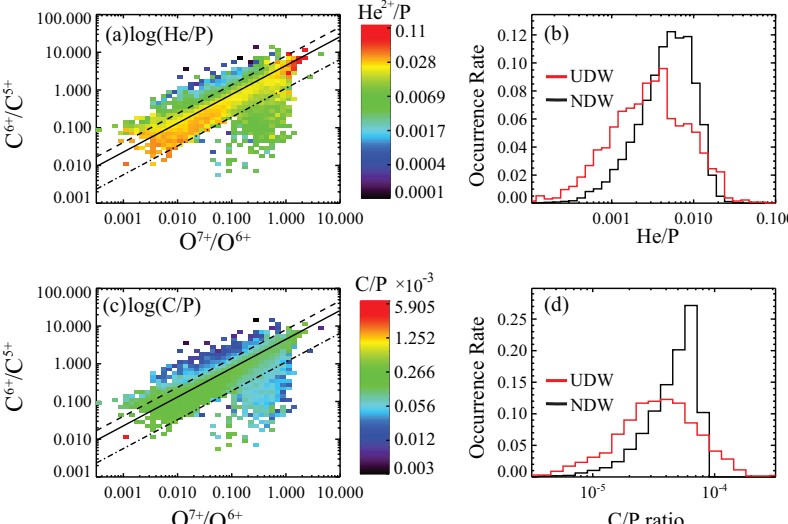

**Figure 1.** (**Left**) distributions of all non-ICME solar wind (January 1998–August 2011). (**a**) He/P ratio and (**c**) C/P ratio in the $\log(C^{6+}/C^{5+})$ and $\log(O^{7+}/O^{6+})$ space. The black solid line is the linear fit of the whole solar wind distribution in 2007, the dashdotted black lines are the threshold by which the Outliers are selected (see Zhao [13] for details). The dashed black lines on the upper-boundary of this plot are one of the thresholds we use to select the UDW in this study. (**Right**) histograms of the He/P ratio (**b**) and C/P ratio (**d**) in the UDW (red) and NDW (black).

## 2. Observations

### 2.1. Data

Located at the Lagrangian L1 point between the Sun and the Earth, the SWICS instrument on board ACE [18] measures the elemental and ion charge state composition of the solar wind heavy ions. In this paper, we use the 2-h measurements by ACE/SWICS from 1998 January to 2011 August. These data have been recalibrated by an improved algorithm [21] which resolves ion species with greater accuracy than before, and which includes charge state distributions of 6 elements (C, O, Ne, Mg, Si and Fe) with uncertainties less than 25%.

We first remove the ICMEs from our database by using the ICME list given by [22], including data 15 h prior to the start time and 6 h after the end of each ICME to minimize the influence of any residual ICME plasma [23]. We also use ACE/MAG (the Magnetic Field Experiment [24]) measurements for the magnetic field and ACE/SWEPAM (the Solar Wind Electron, Proton, and Alpha Monitor [25]) measurements for the proton plasma properties.

### 2.2. Definition of UDW and NDW

Figure 1 shows the elemental abundance ratios of Helium and Carbon over the proton density, He/P and C/P, respectively, in the $O^{7+}/O^{6+}$ versus $C^{6+}/C^{5+}$ ratio frame (O-C frame), for the years from 1998 to 2011. There are three features of this figure that are worth noticing. First, as expected, the two charge state ratios, $O^{7+}/O^{6+}$ versus $C^{6+}/C^{5+}$ are highly correlated for the majority of the data. This correlation can be described through a linear relation:

$$\log(C^{6+}/C^{5+}) = a_1 \times \log(O^{7+}/O^{6+}) + b_1 \qquad (1)$$

We have fitted $a_1$ and $b_1$ parameters for each individual year and listed them in the Table 1 of Zhao [13], along with the correlation coefficients $R_1$. Second, besides the mainstream linear correlation in the O-C frame, there is a subset of solar wind located at

the right-bottom corner of the O-C plot and far away from the mainstream correlation, the so-called "Outliers". These Outliers as selected by the dashdotted blue lines in Figure 1 were investigated by previous studies and found to be active-region or small-flux-rope related winds [13,20]. There are also a substantial fraction of the Outliers that are found to be associated with ICMEs [19], however, these ICME Outliers are not included in this plot because all of the ICMEs are already removed from our study. Last but not the least, we notice that a stripe of data located on the top boundary of the distribution whose He/P and C/P ratios are both dramatically depleted compared to the data in the mainstream correlated distribution. Because these winds are located at the upper boundary of the O-C plot and are characterized as depletion in the Helium and Carbon abundance, we name these winds as "Upper-Depletion-Wind (UDW)". In this study, we focus on these peculiar subset of solar wind, analyze its in-situ properties and investigate its coronal origins. The dashed black lines in Figure 1 are one of the thresholds we use to select these UDWs.

Figure 2 is a review of the O-C plot for each year from 1998 to 2011. Among these correlations, we find that the year of 2007 has the highest correlation coefficient ($R_1 = 0.9$), probably because during that year, the Sun was about to experience the solar minimum and the influence from the Outliers whose origins are associated with active regions are minimal. We use the linear correlation of this year to rule out the upper boundary solar wind by defining the straight-line threshold in the log-log space given by:

$$\log(C^{6+}/C^{5+}) = a_1(2007) \times \log(O^{7+}/O^{6+}) + b_1(2007) + 0.256 \tag{2}$$

This is a upward shift of the linear fitting straight line for the data of 2007, where $a_1 = 0.765, b_1 = 0.649$ by a distance of 0.256 in the O-C frame, which is shown as dashed blue lines in Figure 2. The shifting constant of 0.256 is empirically chosen in order to best separate the UDW from the mainstream distribution. We find that about 80% of the data that are located above this threshold are slow-speed solar wind ($V < 500$ km s$^{-1}$), about 14% of them are fast speed wind ($V > 500$ km s$^{-1}$) and about 6% belong to ICMEs and the buffer period (15 h prior and 6 h after). Since the majority of the wind above this threshold is slow wind, we define the UDW as:

1.  Outside ICME and buffer time
2.  Above the threshold of Equation (2) on the O-C plot
3.  Proton speed less than 500 km s$^{-1}$

In order to understand why these UDWs are located at the upper boundary of the O-C plot, we compare them with the slow-speed solar wind that are located in the mainstream distribution of the O-C plot and possess the similar depletion in C/P ratio as the UDW wind. We define the "Normal Depletion Wind (NDW)" as:

1.  Outside ICME and buffer time
2.  Located between the thresholds of Equations (1) and (2) on the O-C plot
3.  Proton speed less than 500 km s$^{-1}$
4.  C/P ratios are less than the average value in UDW plus one sigma, which is $8.62 \times 10^{-5}$

Note that the purpose of the criteria (4), requiring that the NDW possesses similar Carbon abundance range with the UDW, is to rule out the possible effects from the depletion of the total elemental abundances, but just to focus on the possible influence from the discrepancies in the individual charge states of Carbon and Oxygen between these two groups of slow wind, because they are the key that why UDW is located at a different place on the O-C plot from NDW.

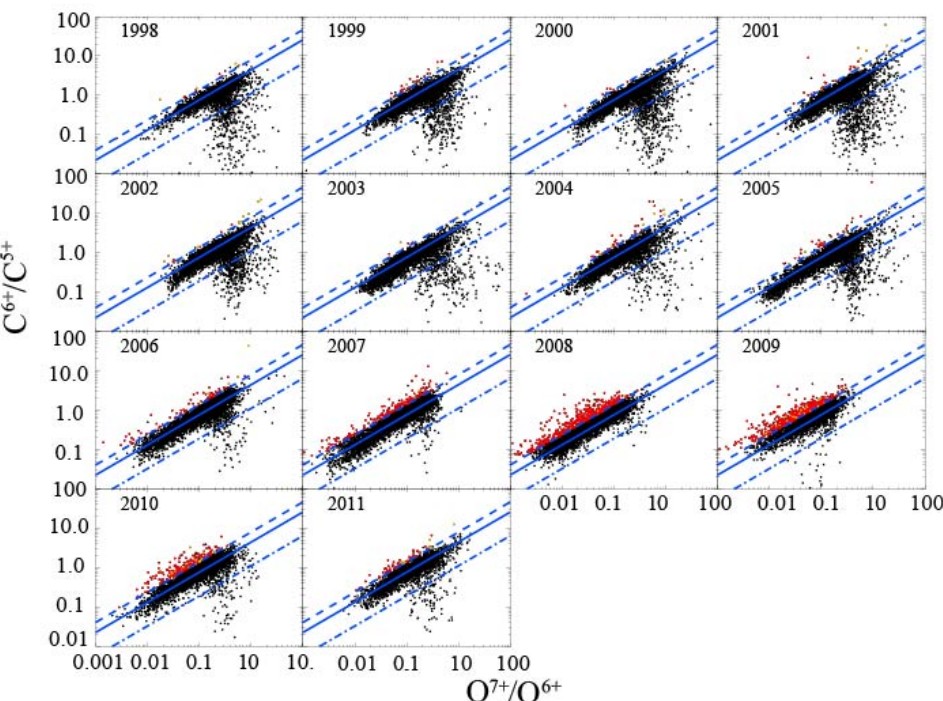

**Figure 2.** Annual comparison of the $O^{7+}/O^{6+}$ and $C^{6+}/C^{5+}$ ratios in solar wind observed by ACE/SWICS from 1998 to 2011. The linear fit of the logarithms of the $O^{7+}/O^{6+}$ and $C^{6+}/C^{5+}$ ratios in 2007 is shown by solid blue lines in each plot; the dashdotted blue lines are the threshold by which the Outliers are selected (see Zhao [13] for details). The dashed blue lines on the upper-boundary of these plots are one of the thresholds we use to select the UDW in this study. The slow-speed ($V < 500$ km s$^{-1}$) UDW is highlighted in red.

## 3. Results

### 3.1. Comparison of UDW and NDW in Elemental Abundances

Figure 1b,d compare the He/P and C/P ratios of the UDW and NDW: both of these two ratios are smaller in UDW than in NDW. Even though the NDWs are required to have C/P ratios lower than the average value of the UDW plus one sigma, the whole distribution of the C/P ratio in the UDW is still left-shifted compared to the NDW wind, which means the values of C/P are still systematically smaller in the former (by $\sim-10\%$). The comparison results are summarized in Table 1.

We also compare the other six elemental abundance ratios over proton density between the UDW and NDW, Mg/P, Si/P, O/P, Ne/P, S/P and Fe/P as illustrated in Figure 3. Overall, all of these elemental abundance ratios are depleted in the UDW compared to the NDW. In addtion to the two ratios of He/P and C/P as shown in Figure 1, the maximum depletion occurs in the Fe/P which is 27.1% lower in the UDW than the NDW; and C/P is the ratio that has the minimum depletion rate among these eight ratios. This is probably because that the NDWs are required to have a similar C/P range as the UDW. It is quite clear that, even though we require the NDW to have a similar C/P range as the UDW, the elemental abundances in the UDW are still depleted by $\sim$10–27%. This implies that the UDW is a subset of slow solar wind that is indeed different from the normal slow solar winds that are located in the mainstream correlated distribution in the O-C plot, and the differences are not caused by the difference in the C/P ratio or any other individual elemental abundance. The detailed average values, standard deviations and depletion rates are listed in Table 1.

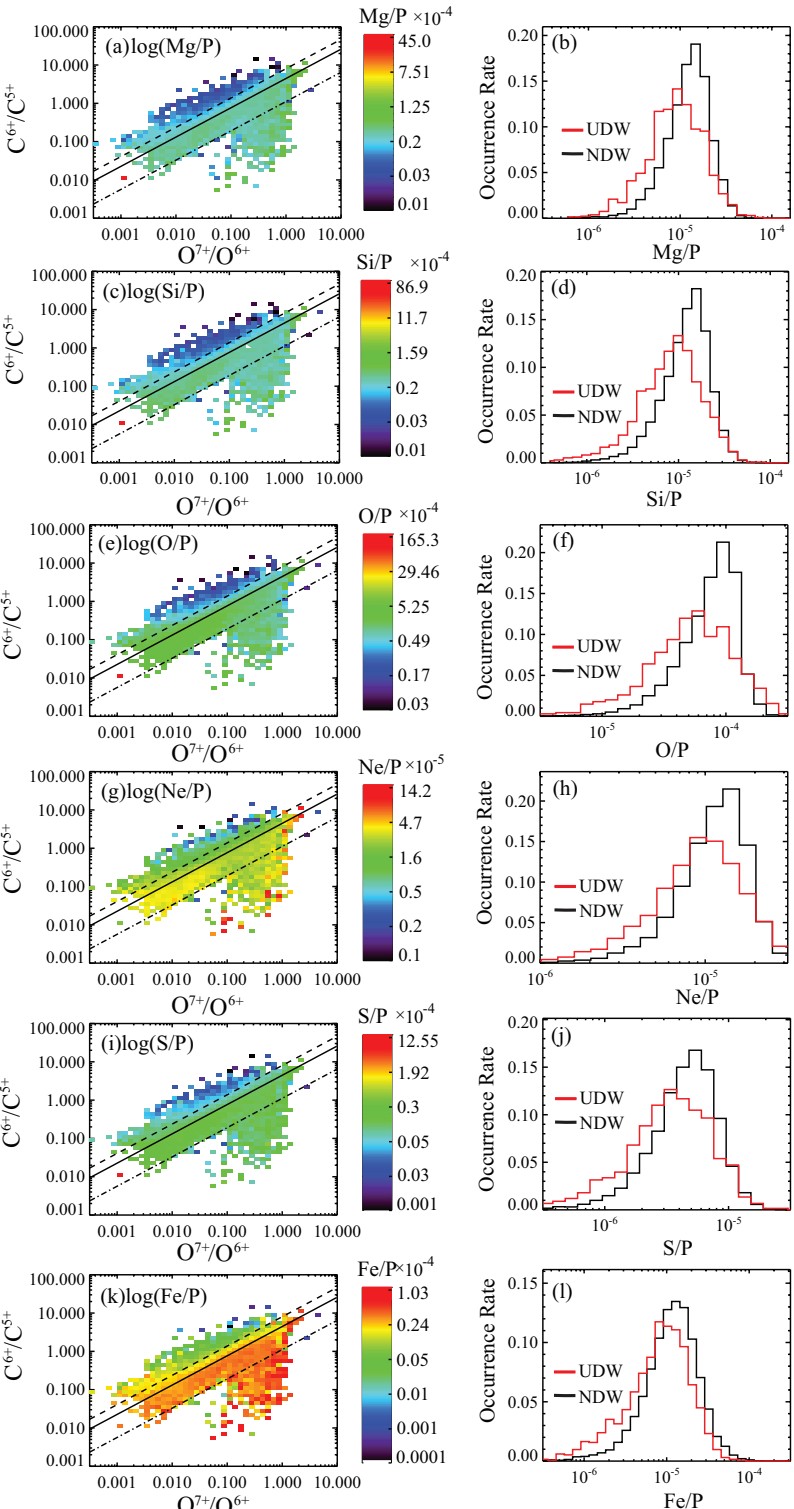

**Figure 3. (Left)** distributions of all non-ICME solar wind (January 1998–August 2011). (**a**) Mg/P, (**c**) Si/P, (**e**) O/P, (**g**) Ne/P, (**i**) S/P, and (**k**) Fe/P in the log(C$^{6+}$/C$^{5+}$) and log(O$^{7+}$/O$^{6+}$) space. The black solid line is the linear fit of the whole solar wind distribution in 2007, the dashdotted black lines are the threshold by which the Outliers are selected (see Zhao [13] for details). The dashed black lines on the upper-boundary of this plot are one of the thresholds we use to select the UDW in this study. (**Right**) histograms of the Mg/P (**b**), Si/P (**d**), O/P (**f**), Ne/P (**h**), S/P (**j**), and Fe/P (**l**) ratios in the UDW (red) and NDW (black).

**Table 1.** Averages, standard deviations, and changing rates of the He/P, C/P, Mg/P, Si/P, O/P, Ne/P, S/P and Fe/P of the UDW and NDW measurements from ACE/SWICS during 1998–2011.

| Mean (Standard Deviation) | UDW | NDW | $\frac{UDW-NDW}{NDW} \times 100\%$ |
|---|---|---|---|
| He/P ($\times 10^{-3}$) | 5.38 (6.23) | 6.5 (4) | $-17.5\%$ |
| C/P ($\times 10^{-5}$) | 4.83 (3.8) | 5.37 (2.0) | $-10.1\%$ |
| Mg/P ($\times 10^{-5}$) | 1.23 (0.88) | 1.55 (0.75) | $-20.8\%$ |
| Si/P ($\times 10^{-5}$) | 1.18 (0.91) | 1.54 (0.83) | $-23.6\%$ |
| O/P ($\times 10^{-5}$) | 7.66 (6.03) | 9.03 (4.11) | $-15.1\%$ |
| Ne/P ($\times 10^{-5}$) | 1.20 (0.76) | 1.36 (0.61) | $-11.5\%$ |
| S/P ($\times 10^{-6}$) | 4.69 (3.53) | 5.69 (3.21) | $-17.6\%$ |
| Fe/P ($\times 10^{-5}$) | 1.18 (0.95) | 1.62 (1.22) | $-27.1\%$ |

*3.2. Comparison of UDW and NDW in Heavy Ion Charge States*

The reason why the UDW is located at the upper boundary of the O-C plot could be either the depletion of its $O^{7+}/O^{6+}$ ratio or the enhancement of the $C^{6+}/C^{5+}$ ratio. In Figure 4, we examine the charge state distributions of Carbon and Oxygen in the UDW and NDW winds. The top panel of Figure 4 shows the distributions of the individual Carbon and Oxygen charge states over the total elemental abundances of their counterparts. For Carbon, it is quite clear that the fractions of $C^{6+}/C$ and $C^{5+}/C$ are very comparable in the UDW and the NDW (Figure 4a). This agrees with the result shown in Figure 4c which illustrates the histograms of the $C^{6+}/C^{5+}$ ratio in these two winds, that are almost overlapped to each other.

Differently, very obvious depletions can be found in the fractions of individual Oxygen charge states of $O^{8+}/O$ and $O^{7+}/O$ in UDW compared to NDW (Figure 4b). Meanwhile, since the $O^{6+}/O$ ratio is basically the same between these two groups of slow wind, the $O^{7+}/O^{6+}$ ratio is strongly decreased in UDW compared to NDW. This is confirmed in the Figure 4d which shows that the histogram of the UDW $O^{7+}/O^{6+}$ is shifted to lower values by about a factor of 3. Table 2 summarizes the mean values, standard deviations and changing rates of these charge states ratios. The $O^{7+}/O^{6+}$ ratio in the UDW is depleted by 64.4%, whereas the $C^{6+}/C^{5+}$ ratio is actually increased by 16.8% compared to the values in the NDW. It is the combined effect of the depletion in the $O^{7+}/O^{6+}$ ratio and enhancement in the $C^{6+}/C^{5+}$ ratio that makes the UDW be located at the upper boundary of the O-C plot and depart away from the correlated mainstream distribution (e.g., Figure 2). In turn, the decrease in the $O^{7+}/O^{6+}$ ratio is almost entirely due to a large depletion in the $O^{7+}/O$ fraction.

**Table 2.** Averages, standard deviations, and changing rates of the $C^{6+}/C$, $C^{5+}/C$, $O^{8+}/O$, $O^{7+}/O$, $O^{6+}/O$, $O^{7+}/O^{6+}$ and $C^{6+}/C^{5+}$ ratios of the UDW and NDW measurements from ACE/SWICS during 1998–2011.

| Mean (Standard Deviation) | UDW | NDW | $\frac{UDW-NDW}{NDW} \times 100\%$ |
|---|---|---|---|
| $C^{6+}/C$ | 0.41 (0.16) | 0.41 (0.14) | 0% |
| $C^{5+}/C$ | 0.39 (0.09) | 0.42 (0.08) | $-8.5\%$ |
| $O^{8+}/O$ | 0.0055 (0.02) | 0.011 (0.03) | $-52.1\%$ |
| $O^{7+}/O$ | 0.060 (0.05) | 0.13 (0.08) | $-54.5\%$ |
| $O^{6+}/O$ | 0.9 (0.06) | 0.83 (0.09) | $8.5\%$ |
| $C^{6+}/C^{5+}$ | 1.38 (2.09) | 1.19 (0.67) | $16.8\%$ |
| $O^{7+}/O^{6+}$ | 0.076 (0.089) | 0.21 (0.17) | $-64.4\%$ |

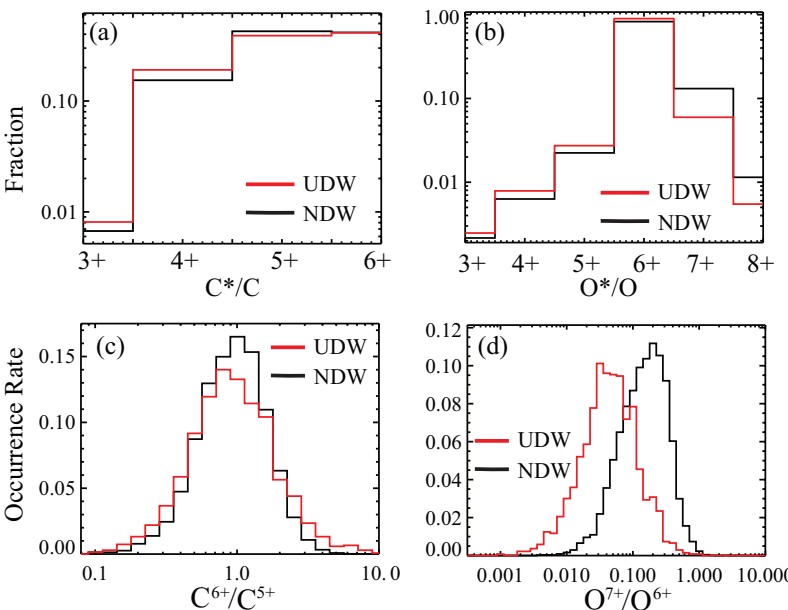

**Figure 4.** (**Top** (**a**,**b**)) Relative distributions of Carbon (Oxygen) charge state over total Carbon (Oxygen) abundance in UDW (red) and NDW (black). (**Bottom**) Histograms of $C^{6+}/C^{5+}$ ratio (**c**) and $O^{7+}/O^{6+}$ ratio (**d**) of UDW (red) and NDW (black).

### 3.3. Comparison of UDW and NDW in Proton Properties and Magnetic Field

Figure 5 compares the proton properties (velocity, density and temperature) and the magnitude of the interplanetary magnetic field ($|B|$) between the UDW and NDW. Because one of the requirements of the UDW and NDW wind is that the proton speed must be less than 500 km/s, these two types of wind show the histograms of the proton speed lower than 500 km/s and the distribution of their speeds are similar (Figure 5a,b). However, the proton density is lower in the UDW than the NDW by about 27% (Figure 5c,d). In addition, the proton temperature is only slightly lower in the UDW compared to the NDW (Figure 5e,f).

The magnetic field measurements, including the three components in the RTN coordinates, $|B_r|$, $B_t$ and $B_n$, and the magnitude of B ($|B|$) are analyzed and compared between these two types of slow winds. We find all of these parameters are decreased in the UDW compared to the NDW. Specifically, $|B_r|$ decreases by 19%, $|B|$, $B_t$ and $B_n$ are all decreased by ~21%. As an example, Figure 5g,h shows the comparison of the magnitude of B. Table 3 summarizes the mean values, standard deviations and changing rates of these parameters.

**Table 3.** Averages, standard deviations, and changing rates of the proton speed (V), density ($N_p$), temperature ($T_p$) and the magnitude of the magnetic field ($|B|$) of the UDW and NDW measurements from ACE/SWEPAM and MAG during 1998–2011.

| Mean (Standard Deviation) | UDW | NDW | $\frac{UDW-NDW}{NDW} \times 100\%$ |
|---|---|---|---|
| V (km s$^{-1}$) | 361.3 (59.2) | 355 (52) | 1.8% |
| $N_p$ (cm$^{-3}$) | 5.85 (3.56) | 8.00 (6.17) | −27.0% |
| $T_P$ (K) ×10$^4$ | 6.96 (4.60) | 7.89 (6.35) | −11.8% |
| $|B|$ (nT) | 4.33 (2.04) | 5.45 (2.89) | −20.6% |

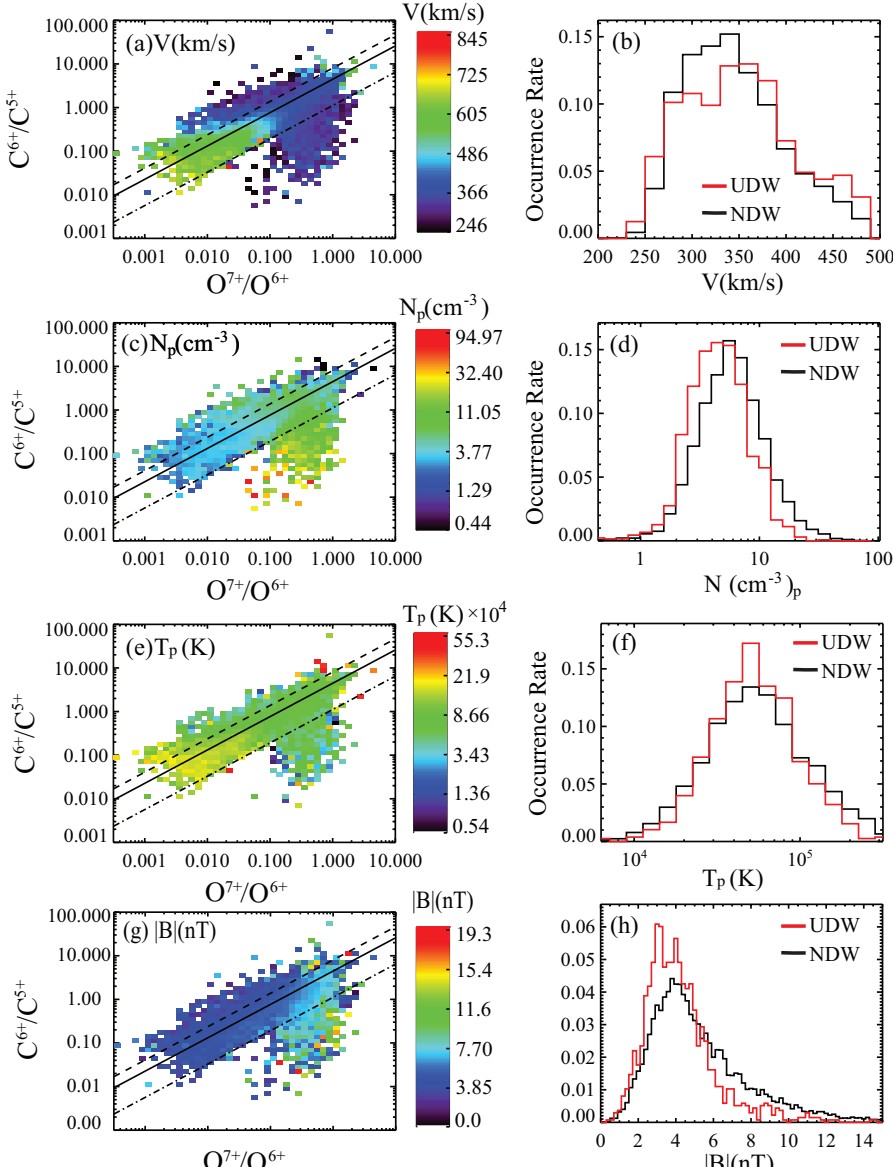

**Figure 5.** (**Left**) distributions of all non-ICME solar wind (January 1998–August 2011). (**a**) proton speed (V), (**c**) density ($N_P$), (**e**) temperature ($T_P$), (**g**) the magnitude of the interplanetary magnetic field ($|B|$), in the log($C^{6+}/C^{5+}$) and log($O^{7+}/O^{6+}$) space. The black solid line is the linear fit of the whole solar wind distribution in 2007, the dashdotted black lines are the threshold by which the Outliers are selected (see Zhao [13] for details). The dashed black lines on the upper-boundary of this plot are one of the thresholds we use to select the UDW in this study. (**Right**) histograms of the V (**b**), $N_P$ (**d**), $T_P$ (**f**), and $|B|$ (**h**) in the UDW (red) and NDW (black).

### 3.4. Solar Cycle Dependence

Figure 6 compares the monthly occurrence rates of the UDW and NDW with the monthly sunspot number (SSN, downloaded from https://www.sidc.be/silso/datafiles (accessed on 1 April 2022)). It is surprising to find that the monthly occurrence rate of the UDW is strongly anti-correlated with the SSN (Pearson and Spearman correlation coefficients are −0.542 and −0.733, respectively, Table 4), with more UDWs observed during solar minimum and less during solar maximum. The monthly occurrence rate of NDW looks also anti-correlated with the SSN, and the Pearson correlation coefficient is −0.545 which is comparable with the value of UDW's, however, its variation from solar maximum to minimum is not as much as the UDW. The fact that the absolute value of the Spearman correlation coefficient between the UDW monthly occurrence rate and the SSN

is much higher than NDW indicates that the UDW has a much stronger dependence with the solar cycle than the NDW.

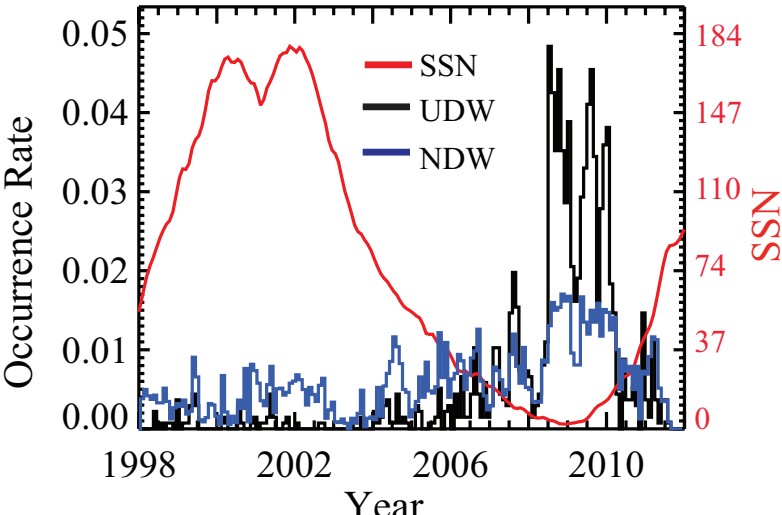

**Figure 6.** Monthly occurrence rates of the UDW (black), NDW (blue) and the SSN(red, dowloaded from https://www.sidc.be/silso/datafiles (accessed on 1 April 2022)).

**Table 4.** Pearson and Spearman correlation coefficient between the monthly occurrence rates of the UDW and NDW with SSN.

| Pearson (Spearman) | UDW | NDW |
| --- | --- | --- |
| SSN | −0.542 (−0.733) | −0.545 (−0.400) |

### 3.5. Coronal Origins

Since the in-situ properties of the UDW are very similar to the NDW, except for the depletion in the heavy ion elemental abundances and $O^{7+}/O^{6+}$ ratio (mainly because of the depletion in $O^{7+}/O$), and proton density, it is useful to examine whether the coronal origins of these two types of slow wind are similar or not, in order to understand what are the causes for the depletion of the heavy ion abundances and $O^{7+}/O$ in the UDW.

We then connect the in-situ observations of the solar wind with their coronal sources using a standard two-step back-mapping technique that can track the ACE measurements to their sources in the solar atmosphere along the magnetic field lines. This technique has been thoroughly described in Zhao [6]. Figure 7 shows the result of the mapping: fractional distributions of the coronal sources of the UDW (magenta) and NDW (blue). We combine the possible coronal origins of these slow wind into four categories: HCS-streamer, active regions and their boundary, quiet Sun region, and coronal hole regions and their boundary. The most different feature of the coronal origins of these two types of slow wind is that more than 50% of the UDW is from quiet Sun regions, whereas equally amounts (∼35%) of the NDW are associated with the active regions and their boundary and the quiet Sun. On the one hand, these different associations with the solar corona explain why the UDW has a stronger solar cycle dependence than the NDW wind. Because the majority of the UDW is originated from the quiet Sun, the monthly occurrence rate of these winds is more strongly anti-correlated to the SSN: the larger the quiet Sun region in the solar minimum, the more frequent the UDW is observed. On the other hand, the association of the majority of the UDW with the quiet Sun also explains why the magnetic field strengths (the three components in RTN and the total magnitude) are slightly lower than those of NDW which is more associated with active regions.

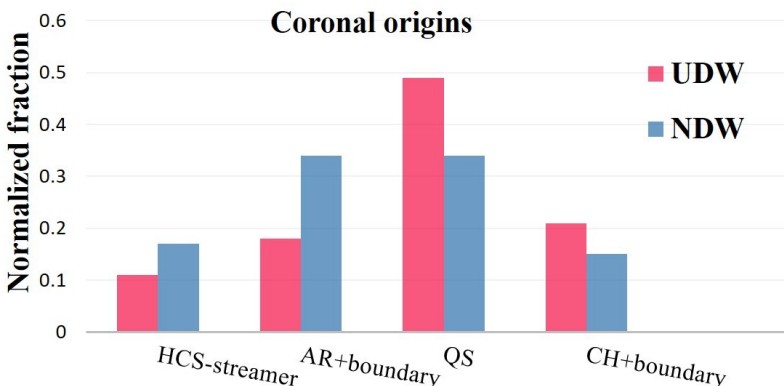

**Figure 7.** Fractional distributions of the UDW (NDW) according to the four combined coronal regions: HCS-streamer, active regions and their boundary , quiet Sun, and coronal holes and their boundary, shown in magenta (blue).

## 4. Summary of the Properties of the UDW and NDW

The main results of the comparison between the UDW and NDW can be summarized as follow:

1. Proton properties: The UDW has similar proton speed, lower proton density and slightly lower proton temperature compared to the NDW.
2. Magnetic properties: All components including the magnitude of UDW are lowered by about 20% compared to the NDW.
3. Composition in UDW:
   (a) The elemental abundance He/P, C/P, Mg/P, Si/P, O/P, Ne/P, S/P, and Fe/P are all decreased by ∼10–30%;
   (b) $C^{6+\sim5+}/C$ and $O^{6+}/O$ are similar as in NDW; whereas $O^{8+\sim7+}/O$ are dramatically decreased, resulting that $C^{6+}/C^{5+}$ is slightly increased but $O^{7+}/O^{6+}$ is greatly decreased.
4. The monthly occurrence rates of UDW is strongly anti-correlated with the SSN.
5. The coronal origins of the UDW are mostly associated with quiet Sun regions; while equally amounts (∼35%) of the NDW are associated with the active regions and their boundary and the quiet Sun.

We find that the UDW is a portion of slow-speed solar wind, possesses peculiar features which are (1) the systematic depletion in the elemental abundance (including proton) and $O^{7+}/O^{6+}$ ratio (caused by the depletion in $O^{7+}/O$) compared to the normal slow-speed solar wind; (2) association with the quiet Sun region and appearance rate is anti-correlated with Sunspot cycle.

## 5. Discussion

The UDW we study in this paper possess a systematically depletion in heavy ion elemental abundances, which is one of the features of the heavy ion dropout events studied by [26]. Weberg [26] reported the first in-situ observations of heavy ion dropouts within the slow solar wind meeasured by ACE/SWICS. They found that these events often exhibit mass-dependent fractionation and are contained in slow, unsteady wind. In order to check if our UDW are related to the dropout events studied by Weberg [26], we check the gravitational settling effect of UDW by investigating the dependence of the depletion rate on the element's atomic mass. Figure 8 presents such a comparison. The depletion rates are the values from Table 1 that we calculated from all of the UDWs and NDWs during 1998–2011. Interestingly, we find that except Helium whose extremely high depletion rate makes it an outlier, the other elements' depletion rates roughly correlated with their atomic mass, indicating that gravitational force may play a role in settling these ions.

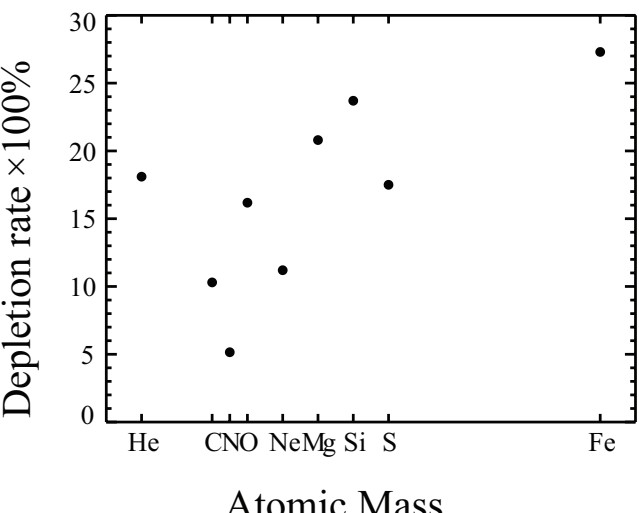

**Figure 8.** Comparison of the depletion rates with the atomic mass of the elements.

However, even if it seems that gravitational force might contribute to the depletion of the heavy ions, we still doubt about the commonality of our UDW with Weberg's dropout events. Because we find that there are several fundamental differences between our UDW and their dropout events which can rule out the possibility that the UDWs are the dropout events that they reported. First, during their dropout events, the $O^{7+}/O^{6+}$ ratios is above 0.1 and is increased compared to the surrounding plasma, e.g., Figure 1 of Weberg [26]. However, the $O^{7+}/O^{6+}$ of UDW is mostly lower than 0.1 (Figure 4) and is not increased but decreased by 64% compared to NDW (Table 2). Second, their dropout events' solar cycle dependence is not very strong, especially for those dropouts that are mass-fractionated. On the contrary, the occurrence rate of the UDW is strongly anti-correlated to the SSN, which is probably caused by their strong association with the quiet Sun region. Therefore, we highly doubt that the UDW studied in this paper is the same slow wind from the dropout events as discussed by Weberg [26].

One of the peculiar features in the composition of the UDW, the depletion of the elemental abundances, will be more apparent if compared with the normal slow solar wind ("NSS", hereafter), which are a subset of slow-speed solar wind defined similarly as NDW but without the fourth criterion, the constrain for the Carbon abundance, as list in Section 2.2. We also compare the properties of the UDW with NSS, and find that except the $C^{6+}/C^{5+}$ ratio, all of the parameters of the UDW, from proton properties to heavy ion composition, are decreased compared to the NSS. Particularly, the heavy ion elemental abundances (He/P, C/P, Mg/P, Si/P, O/P, Ne/P, S/P, and Fe/P) and charge states ($O^{7+\sim8+}/O$, and $O^{7+}/O^{6+}$) are depleted by more than 54% compared to the NSS. However, differently, the decreasing rate of the $O^{7+}/O^{6+}$ ratio in UDW compared to NSS is actually smaller ($-59.5$%) than the rate ($-64.4$%) compared to NDW; and the increasing rate of the $C^{6+}/C^{5+}$ ratio in UDW compared to the NSS is lifted to 28.4% (from 16.8% compared to NDW).

Since it is the $O^{7+}/O^{6+}$ ratio whose depletion makes the UDW displaced on the upper boundary of the O-C plot, the smaller depletion rate of the $O^{7+}/O^{6+}$ ratio in the comparison between the UDW and NSS means that the dramatic depletions (by more than 54%) of the heavy ion elemental abundances do not help the displacement of the UDW from the mainstream of the O-C plot to the upper boundary. The key driver for the displacement of these UDWs must be in the charge state ratios of $O^{7+}/O^{6+}$ and $C^{6+}/C^{5+}$, or in particular, the charge state of $O^{7+}$ (Figure 4 and Table 2). We will report the details of the comparison between the UDW and NSS winds and discuss its implication in a different paper.

## 6. Conclusions

In this work, we have studied the the UDW, a subset of slow solar winds that are located at the upper boundary of the O-C plot and show peculiar heavy ion composition features. We compare the characteristics of these UDWs to the NDWs which are located in the mainstream distribution of the O-C plot and have similar range of Carbon abundance. We find that all of the ratios of the heavy ion abundances over proton (He/P, C/P, O/P, Ne/P, Si/P, S/P, Mg/P, and Fe/P) are depleted by $\sim$11–24% in the UDW compared to the NDW. In addition, the charge states of $O^{7+,8+}/O$ dramatically decreases by more than 50% compared to the NDW, resulting in a large depletion in $O^{7+}/O^{6+}$ ratio (by 64.4%), which is the key cause that makes the UDW displaced on the upper boundary of the O-C plot. The $C^{5+}/C$ and $O^{6+}/O$ also changes a little bit, but they do not affect the displacement of the UDW on the O-C plot too much.

We also compare the coronal origins of the UDW with the NDW and find that the UDW is more likely associated with Quiet Sun regions, whereas equally amounts ($\sim$35%) of the NDW are associated with the active regions and their boundary and the quiet Sun. The differences in the coronal origins could explain some of the features of the UDW, such as its occurrence rate is anti-correlated with the SSN, the proton density and magnetic field in these winds are smaller than the NDW.

However, the physical cause for the huge depletion in the $O^{7+}/O^{6+}$ ratio (mainly due to the depletion of $O^{7+}/O$) is still not clear. Future study including examining more in-situ properties of the wind, such as electron heat flux, proton temperature and entropy, Alv*é*nicity, and remote sensing analysis including examining the electron temperature and density in the wind's source regions will be helpful in understanding the acceleration mechanism and the cause for the anomalous composition of these UDWs.

**Author Contributions:** The authors' individual contributions are: Conceptualization, Investigation, Methodology and Writing—original draft, L.Z. and E.L.; Writing—review and editing, S.T.L. and D.C. All authors have read and agreed to the published version of the manuscript.

**Funding:** The authors acknowledge support from NASA LWS grant 80NSSC22K1015, HGI grants 80NSSC21K0579, 80NSSC18K0647, and 80NSSC18K0645, and HSR grant 80NSSC18K1553. Additionally, S.T.L. acknowledges support from NASA grant 80NSSC20K0192. E.L. acknowledges NASA grants 80NSSC20K0185, 80NSSC21K0579 and 80NSSC22K0750. The authors thank the ACE mission team for making the in-situ magnetic field, plasma, and composition data.

**Institutional Review Board Statement:** Not applicable.

**Informed Consent Statement:** Not applicable.

**Data Availability Statement:** All the data analyzed in this study are publicly available at https://izw1.caltech.edu/ACE/ASC/index.html (accessed on 1 February 2022).

**Conflicts of Interest:** The authors declare no conflict of interest.

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
