# Peer review of "Depletion of Heavy Ion Abundances in Slow Solar Wind and Its Association with Quiet Sun Regions"

_universe, doi:10.3390/universe8080393_

Round 1
Reviewer 1 Report
The origins of slow solar wind have been under active study and debate. This paper focuses on a subset of the slow solar wind with peculiar heavy-ion compositions. The authors showed that the so-called Upper Depleted Wind (UDW) has lower proton properties, a weaker magnetic field, and depleted heavy-ion abundance than the normal-depletion-wind (NDW). They found the primary driver of UDW is the depleted oxygen7+ and the origin of UDW is most likely the quite-sun regions. The authors also discussed the physical cause of such slow wind and admitted that it needs further studies to model the unique oxygen7+/oxygen6+ features. These conclusions are well supported by the data analysis, and the paper is well written.
Author Response
We appreciate the reviewer’s time and effort in reviewing our manuscript and making comments. We followed the reviewer’s suggestion and carefully checked the language of the manuscript. We found a few minor mistakes and typos and have all of them corrected. We also polished the English to make it read better.
Reviewer 2 Report
The paper describes a subset of slow solar winds that are located at the upper boundary of the O-C plot and show peculiar heavy ion composition features. It is interesting and well-written. Please see my major and minor comments below.
1. Figure 4: What do the histograms of Carbon and Oxygen charge state over proton density in UDW and NDW in panels (a) and (b) represent? As you suggest, the reason why the UDW is located at the upper boundary of the O-C plot could be either the depletion of its O7+/O6+ ratio or the enhancement of the C6+/C5+ ratio. So I believe panels (c) and (d) are sufficient. If you want to be more specific, panels (a) and (b) should show the distribution of the Carbon and Oxygen charge states.
Furthermore, there is no relationship between panels (a)-(b) and panels (c)-(d). Some statements in section 3.2, such as " However, the difference of the depletion rates of C6+/P, C5+/P in UDW compared to NDW are very similar, so that the C6+/C5+ ratio in the UDW and NDW wind is also similar", are perplexing. The proton density at solar wind with a mean carbon charge state of 5 (C5+) differs from the proton density at solar wind with a mean carbon charge state of 6 (C6+), so the logical relationship mentioned here, in my opinion, does not hold.
2. Section 3.5: The occurrence rate of NDW is anti-correlated with the SSN, but the source regions of more than half of the NDW are active regions and their boundaries. This may sound like a contradiction.
3. Line 145: “about 6% belong to ICMEs”. Isn’t it the distribution of the non-ICME solar wind, as indicated in the caption of Figure 2?
4. Line 149, I guess you want to say above the threshold of equation (2) here.
5. Figure 5: Please also show the comparisons of B total here.
6. There are also some minor corrections that should be made:
1) There are some “UNW” in the manuscript, please change to “UDW”
2) Line 38 “holes” -> “holes”
3) Figure 4 “oven”-> “over”
4) Figure 5 “ame” -> “Same”
5) Line 230 “0.545” -> “-0.545”
Author Response
We appreciate the reviewer’s time and effort in reviewing our manuscript and making the very helpful suggestions. We answer the reviewer’s questions and update the manuscript as described in the attached PDF file.

Author Response
Response to Reviewer 3
We appreciate the reviewer’s time and effort in reviewing our manuscript and making the very helpful suggestions. We answer the reviewer’s questions and update the manuscript as described below.
- P 1, Line 23: "...accelerated, heated and ionized". = better "... heated, ionized and accelerated"; ´accelerated´ is the last lag for the solar wind creation.
We totally agree with the reviewer’s suggestion, which makes the text much better. We revised the manuscript as suggested.
- P2 Line 50: We changed the text to “with the height above the Sun”.
- P 2, Line 53: If the height will be used instead of ´distance´, than " distance range" = "height range"
We changed the “distance range” to “height range”.
- Figure 1. Unify writing "He/P" below the figure (b) and "He2+/P" in the figure caption.
For the sake of simplicity, we use “He/P” instead of “He2+/P”, and the scientific meaning is not changed because that most of the Helium are He2+.
- P 4, Lines 134-135, the solar minimum was in 2008-2009, so the year is very close to the solar minimum.
We agree. We revised the text as: “probably because during that year, the Sun was about to experience the solar minimum”
- P9, Line 230, 0.545 or -0.545 as written in Table 4, the last column? What is correct?
Yes, it is a typo. We corrected “0.545” to “-0.545”. Thank you!!!
Round 2
Reviewer 2 Report
I'd like to thank the authors for carefully considering my suggestions.
There is only one minor revision that should be made before I recommend that the manuscript be published. You stated in lines 140-141 that the dashed lines are shifted upward by a distance of 0.245, but the distance appears to be 0.256 in equation (2). Please confirm it once more.
Author Response
We appreciate the referee's very careful check through the manuscript. The numbers (0.245) in line 140-141 are typos and we have corrected them to the right ones: 0.256, which is consistent with Equation (2). The revised version of the manuscript is attached.
